# Numerical Modeling of Fire Resistance Test as a Tool to Design Lightweight Marine Fire Doors: A Preliminary Study

**Giada Kyaw Oo D'Amore** [1,*], **Alberto Marinò** [1]  **and Jan Kašpar** [2]

1    Department of Engineering and Architecture, University of Trieste, I-34127 Trieste, Italy; marino@units.it
2    Department of Chemical and Pharmaceutical Sciences, University of Trieste, I-34127 Trieste, Italy; kaspar@units.it
*    Correspondence: giada.kyawood'amore@phd.units.it

**Abstract:** Finite element analysis (FEA) is employed to simulate the thermo-resistance of a marine fire-proof door in the fire-resistance test defined by the International Code for the Application of Fire Test Procedures (2010 FTP Code) and required by the International Maritime Organization (IMO) for marine applications. The appropriate type of simulation adopted (i.e., steady or unsteady) is discriminated on the basis of a comparison between the numerical results and the experimental data. This appropriate model is used to evaluate the critical parameters affecting fire-proof door performance. A remarkable role of the thermal bridge at the door edges in fire resistance is assessed, along with the parameters that allow its reduction. These findings provide insight into how to design a thinner and lighter fire door.

**Keywords:** fire-proof door; fire-resistance test; thermo-mechanical finite element analysis; thermal bridge

## 1. Introduction

Fire safety is one of the most important issues when on board ships. Fire doors are used for slowing or stopping fire propagation and represent a key safety measure, which must be preventively tested and classified [1]. For this purpose, International Convention for the Safety of Life at Sea (SOLAS) amendments, within the provisions on fire safety on board, made effective on 1 July 2012 the International Code for the Application of Fire Test Procedures (2010 FTP Code), a mandatory safety fire test [2]. In this test, a door is exposed to a prescribed time-temperature heating schedule, depending on the fire-door class, and both mechanical and thermal limits are imposed to assess the door performance [1].

The design of conventional fire doors is based on well-established methodologies, aimed at optimizing their thermal behavior in order to comply with the requirements of the fire-resistance test. However, the thermal gradients may induce significant distortions: the door tends to bend away from its supporting frame due to non-uniform temperature distribution that may lead to flame and smoke propagation, resulting in fire-test failure.

The performance of fire-proof doors in the fire tests has been addressed both experimentally and using numerical modeling [3–8]: these simulations were typically aimed at reproducing the experimental results and assessing the computational methods. However, Izydorczyk et al. [9] highlighted the variability of the results after testing the effects of several fire-door construction parameters and materials on the fire-resistance performance.

Alternative ways to improve the fire resistance have also been demonstrated: Kong et al. [10] used pressurized water inside the door to maintain constant temperature during the fire test, thus avoiding

significant deformations. Moro et al. [11] proposed an innovative design scheme, where the mechanical response of the door is enhanced by placing the insulating material on the outer sides of the door, rather than in the inner part. Nevertheless, this solution does not permit the use of rock wool, the most widely used insulator due to its low cost, non-flammability, and lightness [12], as it is too soft to be placed outside of the door. In this case, higher density insulators and adequate surface coatings must be used [11].

The present study aims to identify a proper finite element method (FEA) model that allows one to identify the origin of the excessive temperature gradients that develop on the door [6], determining high distortions. Use of numerical models saves time and costs during the door design phase by reducing the number of prototypes to be constructed and tested. The FTP procedure is indeed an expensive test. The modeling provided useful indications, which allowed the decrease of temperatures on the fire-unexposed door surface without modifying or making the production process complicated.

Thus, a thinner and lighter door could be designed on the basis of the obtained results and its fire resistance simulated using the FEM model developed.

## 2. Materials and Methods

### 2.1. Numerical Modeling of the Fire Resistance Test

#### 2.1.1. Fire-Proof Door Characteristics and Test Conditions

A single-leaf fire-proof marine door was analyzed. The door had a clear opening of 1000 mm and a height of 2000 mm. The leaf (1117 mm width and 2117 mm height) used carbon steel sheets 5 mm thick for the edges and the fire-unexposed side and 1.5 mm thick for the fire-exposed side. The insulating material (ROCKWOOL® 251) was 60 mm thick. The door frame, made of steel profiles having thickness 5 mm, was screwed on the bulkhead by screws with a wheelbase of 150 mm. The door was fixed to the frame by means of three hinges on one edge, while a lock was arranged on the opposite edge (Figure 1a).

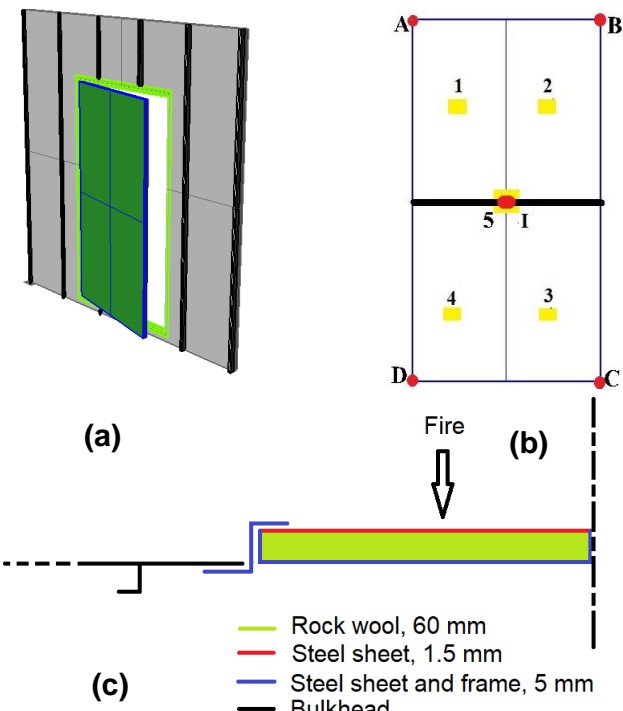

**Figure 1.** (**a**) Door model geometry; (**b**) test measuring points: thermocouples location (1, 2, 3, 4, 5) and displacements measuring points (A, B, C, D, I); (**c**) door section with thickness of the components specified.

In the FTP Code fire test, the door is heated from room temperature up to 945 °C with a prescribed time-temperature relation shown in Equation (1):

$$T = 345 \log_{10}(8t + 1) + T_0 \qquad\qquad (1)$$

where t is the time (min) and $T_0$ the room temperature (°C). In our case the room temperature was 28 °C.

For an A-60 class door, considered here, the fire test lasts 60 min. Temperatures and displacements must be monitored in five positions (Figure 1b); on the fire-unexposed side (i) the final temperature of each thermocouple must not exceed 180 °C above $T_0$ in, and (ii) the medium temperature of the five thermocouples must not exceed 140 °C above $T_0$. Moreover, only small gaps between the door and the frame are tolerated to ensure that no flame passes through the door (detailed experimental procedure and limit parameters are reported in the FTP Code [1]).

### 2.1.2. FEM Modeling

The modeling of the fire test was performed with the software MSC Nastran with Patran 2017, employing two steps: firstly, a thermal analysis to evaluate the temperature distribution, then a structural analysis based on the previous thermal results. The two analyses can be considered uncoupled as the thermal distortions did not significantly affect the temperature distribution during the test [11]. Both analyses are non-linear as the materials' properties depend on temperature. The thermal analysis was performed both in unsteady- and steady-state conditions, while the structural analysis was performed only in steady-state conditions: the key constituent materials of the fire door were not viscoelastic; thus, the deformation history did not significantly influence the final deformations at the maximum temperature [7].

The FEM model has been validated by a comparison with the RINA (Italian Naval Register) certification [13] of the fire test carried out on the aforementioned door.

The geometry has been discretized (Figure 2) using hexahedral elements (3D) for the insulating material, and quadrilateral (2D) elements for components that have thickness negligible with respect to the other dimensions (e.g., door sheets, frame, bulkhead, and stiffeners). In the literature, it has already been shown that 2D elements, in addition to being computationally more efficient [7], give the same results as 3D elements [14].

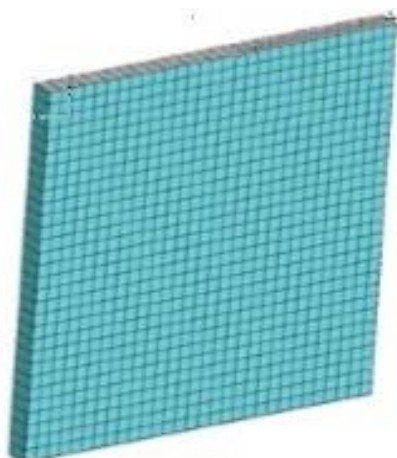

**Figure 2.** Detail of the door's mesh.

The model was bound by fixing rotations and translations at the edges of the bulkhead, since it was welded to the test machines. R-type elements were used to model both the lock and the hinges: in particular, RBE2 elements, that are multi-point constraints (MPCs), were adopted. These connectors are rigid elements that force the nodes to have the same displacements: they provide independent

degrees of freedom (DOFs) at a grid point and dependent DOFs at one or more grid points. The thermal elongation of steel was adopted for the connectors.

The following conditions were employed on the door fire-exposed side: (i) for the steady-state analysis, a fixed temperature equal to 945 °C (that is, the maximum temperature reached after 60 min) was imposed; (ii) for the unsteady-state analysis, radiation and convection were considered constant and the temperature gradient followed the standard time-temperature curve according to Equation (1) [1]. The natural convection coefficient ($h_c$) was set to 25.0 W $(m^2 K)^{-1}$ [15], and uniform radiation heat flow was assumed with a view factor set to 1.0 [7], adopting the black-body condition [15].

For the door fire-unexposed side, the following conditions were employed: (i) for the steady-state analysis, convection ($h_c$) and radiation ($h_r$) coefficients were set to 4 and 6 W $(m^2 K)^{-1}$, respectively; (ii) for the unsteady-state analysis, the steel emissivity ($\varepsilon$) was set to 0.3, and $h_c$ was maintained at equal to 4 W $(m^2 K)^{-1}$ [6,15]. The emissivity factor was set to 0.3 in accordance with the temperatures reached on the door fire-unexposed side without considering radiation and convection: the value of 0.7, recommended by the Eurocode 1 [15], is justified for temperatures higher than 520 °C. In this study the relation reported in Sadiq et al. [16] (Equation (2)) for the temperature range 380–520 °C was adopted.

$$\varepsilon = 0.00293T - 0.833 \tag{2}$$

The time step of the unsteady analysis was defined using an adaptive method: setting as input an initial time step, the software automatically calculated the subsequent time steps based on the convergence conditions. An initial time step of 10 s provided stability and accuracy for the integration process when the above-mentioned software was used.

*2.2. Material Properties*

For carbon steel (Fe 37/360), thermal and mechanical properties needed for the numerical analysis were tabulated on the materials datasheet for ambient temperature (Table 1) and integrated with data available in the literature for high temperatures [17].

**Table 1.** Material properties at 20 °C.

| Material | Young's Modulus E (GPa) | Poisson's Ratio $\nu$ (-) | Density $\rho$ (kg m$^{-3}$) | Thermal Conductivity $k$ (W (m K)$^{-1}$) | Thickness (mm) |
|---|---|---|---|---|---|
| Carbon steel | 190 | 0.3 | 7800 | 27.29 | 5.0–1.5 |
| Rock wool | $4 \times 10^{-6}$ | 0 | 180 | 0.035 | 60 |

For the insulating material, the temperature dependence was considered only for the thermal conductivity; the insulating material had no significant structural contribution [7], so the mechanical properties were considered constant for simplicity. The rock-wool properties as reported in the datasheet (Table 1) were integrated with experimental measurements.

To account for the influence of the temperature, thermal conductivity in the range 20–80 °C was measured with a Heat Flow Meter-HFM 446 Lambda Small (NETZSCH Group, Selb, Germany), according to the ASTM C518 [18], while temperatures up to 200 °C were measured using the constant heat flow method [19], with a specifically modified muffle. The conductivity values for temperatures higher than 200 °C (k (T)) were calculated according to the ISO 10456:2007 [20], using Equation (3):

$$k (T) = k_{ref} \cdot \exp(f_T (T_i - T_{ref})) \tag{3}$$

where $k_{ref}$ is the value of the thermal conductivity in W (m K)$^{-1}$ at a reference temperature $T_{ref}$ in °C (20 °C in our case), $T_i$ is the temperature of interest in °C, and $f_T$ is a scale factor tabulated in the standard ISO (in this study a factor for glassy materials $f_T$ = 0.003 was adopted). This assumption worked properly for the insulator used in this work, but it should be remembered that thermal

conductivity of insulating materials at high temperatures can be affected by different parameters (e.g., density, presence of binder) [21,22].

## 3. Results

With the aim of assessing a proper FEM model, the comparison between steady- and unsteady-state thermal analysis is addressed in this section. Steady-state analysis is generally adopted in the literature, as it allows one to reduce the computational burden [6,11]. The results of the comparison were assessed using the experimental data in a report on a FTP test performed by RINA [13].

The unsteady-state FEM model was then applied to evaluate the effects of the thermal bridge and, based on the findings, a thinner and lighter fire door was designed. The assessed numerical model was finally used to evaluate the thermal performances of the new door.

However, before addressing these results, we focus the fact that the door and the frame can be considered thermally uncoupled, consistent with the literature [14]. Figure 3c shows that the effects of the frame are visible only in close proximity to the door border and there are no effects on the temperatures at the measuring points (Figure 1b) situated on x-coordinates of 0.25 (points 1, 4), 0.50 (point 5), and 0.75 (points 2, 3), as required by the FTP Code [1]. Consequently, thermal data reported in the next sections are related to the door-only model (Figure 3b) that allows one to perform faster and easier simulations. Structural analysis was carried out using the entire model (Figure 3a), as the distortions were typically manifested at the door borders where the highest thermal effects of the frame-door coupling are seen.

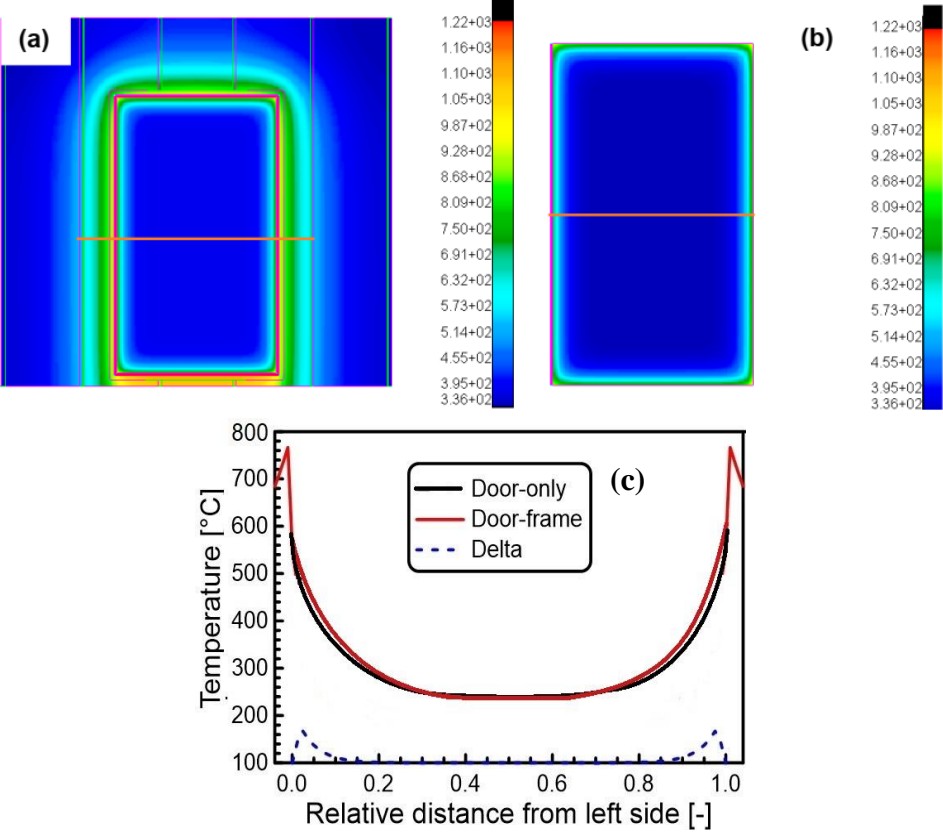

**Figure 3.** Temperature distribution (K) on the door fire-unexposed side calculated at the final stage of the fire test (60 min): (**a**) door-frame and (**b**) door-only model; (**c**) transversal temperature calculated along the horizontal lines in (**a**) and (**b**) (0 and 1 represent the door edges). Delta: temperature difference between the two profiles. Analysis performed using unsteady finite element method (FEM) analysis (compare next section).

### 3.1. Steady-State vs. Unsteady-State Thermal Analysis

Results obtained from the unsteady-state thermal analysis were compared with the experimental data obtained in the fire-resistance test certification to assess the numerical model [13]. In Figure 4 the theoretical heating time-temperature curve imposed by the FTP Code (Equation (1)) is compared with both the calculated and the experimental temperatures obtained on the door fire-exposed side.

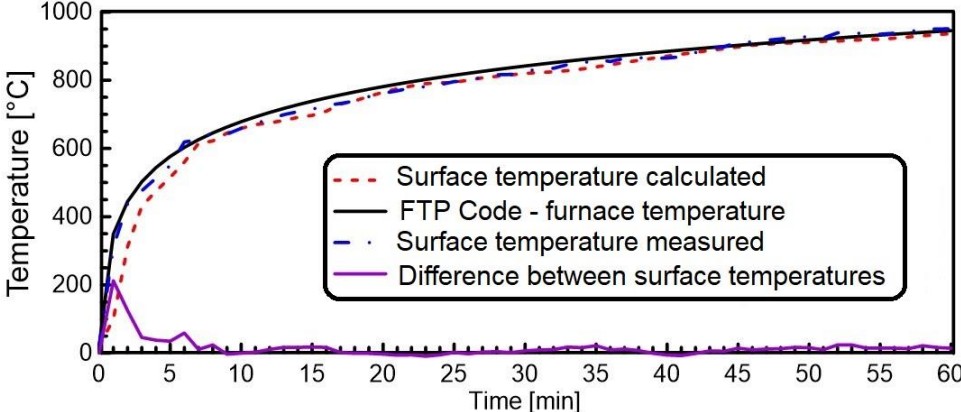

**Figure 4.** Temperature profiles on the door fire-exposed side. Delta: difference between measured and modeled surface temperatures.

Except for the first 8 min of the test, where the calculated curve features a significant delay in the temperature rise compared to the experimental one (differences up to 212 °C), a good agreement was observed as the temperature differences never exceeded 2%. Importantly, the final temperatures, after 60 min, were nearly coincident (945 ± 5 °C, equal also to the temperature used for the steady-state analysis).

On the door fire-unexposed side, a transversal temperature profile was calculated along the line passing through the measuring point 5 (see the black line in Figure 1b) using both unsteady- and steady-state analysis (Figure 5).

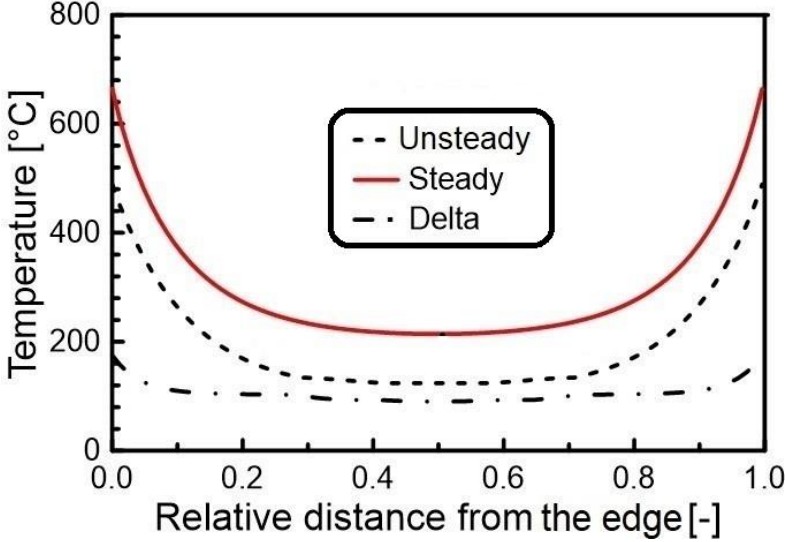

**Figure 5.** Transversal temperature profiles calculated at the final stage of the fire test (60 min) using steady- and unsteady-state FEM methodologies. Delta: temperature difference between the two analyses.

There is a striking difference in the temperatures obtained by the two calculation methodologies: the steady-state analysis significantly overestimated the temperatures; using this model, the door would not meet the FTP limits (not to exceed 200 °C in each measuring point in this study). This is clearly confirmed by the data reported in Table 2, which show that the unsteady-sate analysis is the only method capable of properly modeling the final temperatures in all the measuring points.

**Table 2.** Comparison of modeled and measured temperatures at the final stage of the fire test (60 min in the measuring points required by the International Code for the Application of Fire Test Procedures (2010 FTP Code) (Figure 1b). Ambient temperature equal to 28 °C in this study.

| Point | T Steady Model (°C) | T Unsteady Model (°C) | T Measured * (°C) |
|---|---|---|---|
| 1 | 248 | 146 | 149 |
| 2 | 248 | 146 | 145 |
| 5 | 214 | 124 | 127 |
| 3 | 248 | 146 | 140 |
| 4 | 248 | 146 | 146 |

* Data obtained from Italian Naval Register (RINA) fire test certification report n° 2005CS015100/1 [13].

As illustrated in the experimental section, the deformation history did not influence the final deformations at the maximum temperature [7]; thus, the structural analysis was carried out under steady-state conditions, using as thermal load the final temperatures obtained with an unsteady-state analysis. As explained above, the entire model (door with frame) was used for these simulations. Also, in this case, except for the measuring point B, a good agreement was found between measured and calculated displacements. The different displacements measured between point B and point C during the FTP test (see Table 3, in B the displacement is more than 4 times greater) can be justified by a heat flow directed upwards. The present numerical model cannot evaluate this effect: the boundary conditions are considered constant as suggested by Eurocode 1 [15] and, the door being symmetric with respect to the x- and z-axes, its thermo-mechanical response is symmetric (displacements equal in points A–D and B–C).

**Table 3.** Comparison between model and measured displacements at the final stage of the fire test (60 min).

| Point | d Model (mm) | d Measured * (mm) |
|---|---|---|
| A | 1.0 | 2.0 |
| B | 4.5 | 13.0 |
| C | 4.5 | 3.0 |
| D | 1.0 | 2.0 |

* Data obtained from RINA fire test certification report n° 2005CS015100/1 [13].

The FEM model consistently overestimated the measured displacements when the thermal load from the steady-state simulations was used (data not reported).

*3.2. Effect of Thermal Bridge*

In order to identify the specific contributions to the thermal load affecting the door fire-unexposed side, the heat flow in measuring point 1 (taken as reference and located at a distance of 312.0 mm and 562.5 mm from the edges of the door along x-axes and z-axes, respectively) was decomposed into the three Cartesian directions (Figure 6). Remarkably, the heat-flow component in x-direction was significantly higher than that transmitted in y-direction. This highlights the fundamental importance of the thermal bridge that originated from the door edges in determining the fire-performance of the door. Consistently with the position of point 1 (farther from the upper door edge than from the side edge), a negligible contribution was observed in the z-direction.

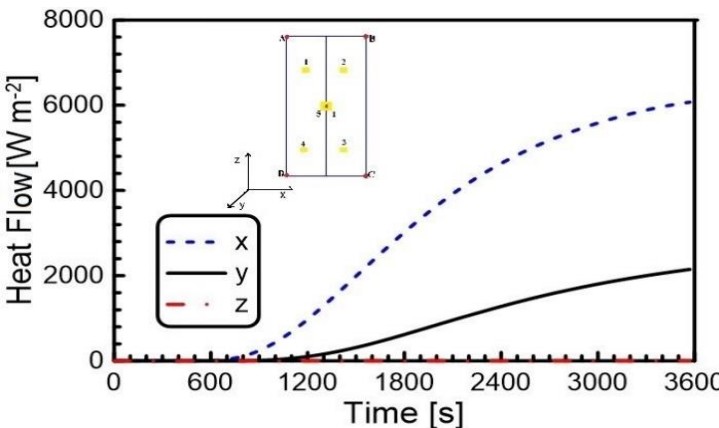

**Figure 6.** Thermal measurement point 1: deconvolution of the heat flow as a function of time along the Cartesian directions (x, y, z).

Given the effect that the thermal bridge had on the door performance, different door configurations were simulated to assess the origin of this effect. These simulations showed that by reducing the thickness of the edge steel sheets, the thermal bridge was significantly reduced. Figure 7 compares the transversal temperature on the fire-unexposed side of the standard door A analyzed so far (edge steel sheets 5.0 mm thick) and of a modified door A having edge steel sheets 1.5 mm thick. Remarkably, by decreasing the thickness of the door edge, a decrease of the temperatures ranging from 20 °C at the center to 130 °C at the edges was calculated.

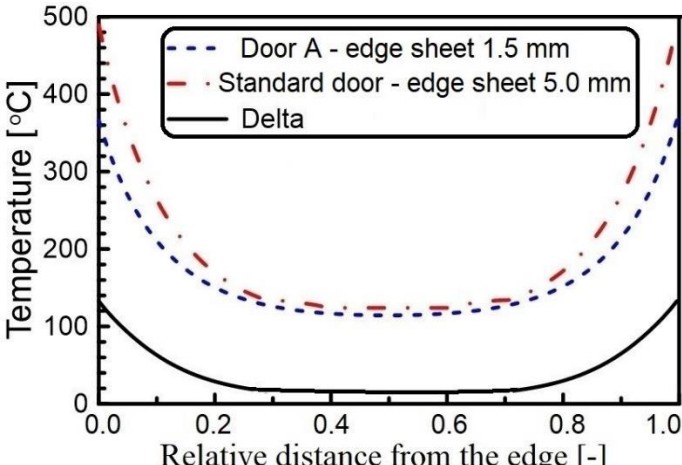

**Figure 7.** Transversal temperature profiles calculated at the final stage of the fire test (60 min) using unsteady-state FEM model: comparison of the standard and modified doors. Delta: temperature difference between the standard and modified doors.

*3.3. Thermo-Resistance Numerical Simulation of a Thinner and Lighter Fire Door*

On the basis of the temperature drop obtained by reducing the edge steel sheets (Figure 7), an advanced door (door B) was designed which was thinner and lighter. The new door had a thickness of 52 mm and carbon steel sheets 1 mm thick.

Temperature profiles obtained by the numerical simulation after 60 min along the line passing through measuring point 5 on the door fire-unexposed side both for door A and door B are compared in Figure 8. Although the temperatures reached by door B (reported in Table 4) are on average higher than door A, they meet the limits imposed by the FTP Code. More interesting is the difference in the shape of the transversal profile, especially the linear part observed for door B, which suggests that the effects of the thermal bridge are limited to a relative distance of 0.2 (Figure 8).

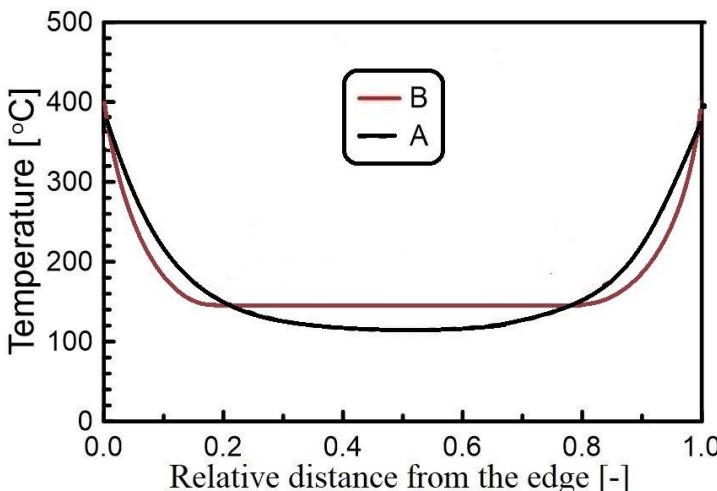

**Figure 8.** Transversal temperature profiles calculated at the final stage of the fire test (60 min) using unsteady-state FEM model: comparison of door A and door B.

**Table 4.** Calculated temperatures at the final stage of the fire test (60 min) on the door A and B. Ambient temperature equal to 28 °C in this study.

| Point | Door A (°C) | Door B (°C) |
|---|---|---|
| 1 | 139 | 148 |
| 2 | 139 | 148 |
| 5 | 119 | 147 |
| 3 | 139 | 148 |
| 4 | 139 | 148 |

Notably, door B weighs 61 kg compared to the 161 kg of door A.

## 4. Discussion

As stated in the introduction, there is need for a reliable modeling of the FTP test to facilitate the design of novel fire-safety devices, such as naval fire doors. This aspect has been addressed in the literature using FEM modeling; however, the use of steady-state analysis has generally been suggested, to also limit the computational burden [6]. This simplification is justified in the literature by the fact that the temperatures on the fire-unexposed side of the door were asymptotic after a few minutes, and the final temperatures calculated with both steady- and unsteady-state analysis were nearly coincident (Figure 4).

These previous findings, however, seem to be related to the specific systems analyzed [6,14], since our results clearly highlight the necessity of the unsteady-state analysis to properly fit the experimental data, both thermal and structural: the steady-state analysis significantly overestimates the temperatures on the door's fire-unexposed side (Section 3.1).

Moreover, the study of the heat fluxes and the evolution of the thermal bridge is not possible without the unsteady-state simulations, while the importance of these phenomena has been evidenced (Section 3.2). The reduction of the computational burden between steady- and unsteady-state analysis is minimal if compared with the cost and time of prototype construction and testing. Thus, to use the simulations as a tool to optimize the fire doors during the design phase, the unsteady-state analysis is strictly necessary.

The most important result of this study is the identification of the critical role of the thermal bridges in decreasing the fire-resistance of the door in the FTP test. Generally, to achieve adequate fire resistance, lower the thermal gradients, lower the door distortions, and consequently lower the

possibility of flame and smoke propagation through the openings between the door and the frame in case of fire.

Importantly, we show that the thermal bridge can be reduced by reducing the thickness of the steel sheet at the door edges (Section 3.2, modified door A). This modification also allows the reduction of the temperature at the center of the door, leading to the possibility of designing lighter and thinner fire doors.

Door B has therefore been designed (Section 3.3) and a numerical analysis has been performed to evaluate its thermal performance. Door B meets the temperature requirements imposed by the FTP Code, making an improvement in terms of weight and size: door B weighs 62% less than door A and is thinner by 22%. Moreover, the temperature profiles reported in Figure 8 infer significant reduction of the thermal bridge effects in door B with respect to door A.

## 5. Conclusions

The preliminary results reported in this paper indicate that a proper modeling of the FTP test (i.e., steady- vs. unsteady-state analysis) allows the identification of the critical factors determining fire resistance and hence the feasibility of a priori design of effective fire-safety measures.

In particular, the origin and fundamental importance of the thermal bridge in determining the thermo-mechanical resistance of the fire door have been highlighted. This allowed the identification of a route to limit the extent of the thermal bridge, leading to a thinner and lighter fire door.

Finally, to create a more reliable numerical model based on a self-consistent methodology, further refinement is necessary, such as use of properly derived convective and radiation coefficients rather than using the standard values reported in the literature [15], effects of steel emissivity at different temperatures [7,8,23], and effects of heat flow upwards as reported in Section 3.1.

**Author Contributions:** G.K.O.D. performed experimental measurements and thermal simulations. G.K.O.D., A.M., and J.K. wrote the paper. A.M. and J.K. overviewed the research and edited the paper. All authors have read and agreed to the published version of the manuscript.

**Funding:** This research received no external funding.

**Acknowledgments:** Materialscan S.r.l. are acknowledged for financial support. Officine Del Bello MBM Srl is acknowledged for the RINA certification data.

**Conflicts of Interest:** The authors declare no conflict of interest.

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
