# Peer review of "Numerical Modeling of Fire Resistance Test as a Tool to Design Lightweight Marine Fire Doors: A Preliminary Study"

_jmse, doi:10.3390/jmse8070520_

Round 1
Reviewer 1 Report
The topic of the paper is interesting but there are some inaccuracies and missing information for the reader for following the text. Before publishing the authors should improve the paper (see remarks in attached document). Especially informations about the assumed high-temperature thermal and mechanical properties is missing and has to be supplemented.

Author Response
Response to Reviewer 1 Comments
Thanks for your comments.
The corrections are here highlighted.
Point 1: Explanation of the term MPC/RBE2 is missing
Response 1: Line 112, explanation of the term MPC/RBE2 is addressed. RBE2 are R-type elements, that are multi-point constraints (MPCs); they are rigid elements that force the nodes to have the same displacements. They provide independent degrees of freedom (DOFs) at a grid point and dependent DOFs at one or more grid points.
Point 2: Eurocode 1 (EN 1991-1-2) recommends a coefficient of 4 W/(m²K) for heat transfer by convection on the unexposed side; for emissivity 0,7 is recommended in EN 1993-1-2. It should be clarified why the values chosen by the authors deviate from the EN-values and what if the results are influenced by these deviations significantly
Response 2: Line 127, we have motivated the use of the emissivity factor equal to 0.3 has been adopted on the basis of the temperatures reached on the fire-unexposed side of the door without considering radiation and convection: the value of 0.7 recommended by the Eurocode 1 is justified for temperatures higher than 520 °C due to the formation of surface oxide. Citations has been addressed. For the convection coefficient, has been chosen the one proposed by the Eurocode 1 and equal to 4 W/m2K, the citation has been addressed. Actually, as the referee recognised, this is an interesting point that can influence not only the results of calculation but also of the effectiveness of the door in the fire test.
Point 3: The thickness of the materials should be listed in Table 1
Response 3: Table 1, the materials thickness has been included.
Point 4: These values should be compared with values for high temperature thermal conductivity published in literature (e. g. Schleifer (2009) and Livikiss (2008))
Response 4: We believe that the quoted reference Schleifer 2009 is related to Vanessa Schleifer’s ETH Zurich PhD thesis entitled “Zum Verhalten von raumabschliessenden mehrschichtigen Holzbauteilen im Brandfall” which effectively discuss the effects of different parameters of thermal conductivity of rockwool including effects of density: see for example the relation on p.68. However, the accessibility of a German text to wide audience is limited and accordingly we prefer to use as citation the paper Frangi, A., Schleifer, V. & Hugi, E. A New Fire Resistant Light Mineral Wool. Fire Technol 48, 733–752 (2012) https://doi.org/10.1007/s10694-010-0209-2 and
Sjöström, J.; Jansson, R. MEASURING THERMAL MATERIAL PROPERTIES FOR STRUCTURAL FIRE ENGINEERING. 2012, 11 that cite the PhD thesis.
As for the Livikiss (2008) reference, we are sorry but searches on either Scopus or Scifinder (CAS) or Web of Science or Google Scholar did not give any result for Livikiss as author.
Line 153, we have included a warning that the thermal conductivity should be carefully considered as could be affected by different parameters.
Point 5: Fire tests on doors (for buildings) show that often the highest temperatures occur near the border of the door leaf adjacent to the frame. Are the measuring points shown in Fig. 1b) corresponding to the requirements in FTP code? When "yes", then this should be mentioned, when "no" this should be discussed.
Response 5: Line 168, we have specified that measuring points correspond to the requirements of the FTP Code.
Point 6: The legend has to be clearly: black line: FTP Code (gas temperature, action) red dotted line: surface temperature calculated, blue dotted line: surface temperature measured, purple line: difference between measured and modelled surface temperature
Response 6: Figure 4, we have changed the legend clarifying the entries, according to the referee’s suggestions.
Point 7: "theoretical FTP" curve" is a gas temperature and exper. temp. is a surface temperature. Be careful by comparing these curves! Because of high thermal conductivity of steel the surface temperature approaches the gas temperature by longer time of exposure
Response 7: Line 190, we have changed the comparison with the modelled temperature on the unexposed side of the door. We deleted the comparison with the FTP curve as the reviewer correctly evidences that the FTP curve is a gas temperature and not a surface temperature as the experimental and the modelled ones. Thank you!
Point 8: It would be helpful to add a drawing of the section of the door profile with dimensions
Response 8: Figure 1, we have addressed the Figure 1c with the door section with the thickness of the components specified.
Point 9: It has to be clearly defined which measuring point is considered (coordinates)
Response 9: Line 208 (caption of Table 2), we have specified that the measuring points as required by the FTP Code are considered.
Point 10: Description of element type in FE analysis is missing. How are the mechanical properties modelled? Only elastic? Which values for Youngs modulus, stress-strain relation and thermal strain are assumed for high temperatures, EN 1993-1-2 values?
The literature shows clearly that modelling elastic properties and neglecting thermal strain leads to a huge underestimation of deformations in fire situation.
Response 10: Line 106, we have addressed the grid element typology used in the FEM analysis (hexahedral for the insulating material and quadrilateral for the other door components). While, in Section 2.2 is specified that data reported in the Eurocode 3 are used for the high temperatures.
Point 11: It has to be precised in the legend which case is presented (sheet 5.0 mm or 1.5 mm)
Respinse 11: Figure 7, the legend has been changed specifying the case presented.
Point 12: That is not true! Fig. 5 and table 2 show a significant difference between steady-state and unsteady simulations
Response 12: Line 276, thank you very much for careful reading: you are perfectly right! The sentence was intended as referred to the literature – not to the present work! This has been now properly specified.
We wish to thank the refereee for careful reading and stimulating comments.
Kind regards.
Reviewer 2 Report
The paper is well written and merits publication. I have only one comment or query. Could the authors elaborate on the discrepancy in the displacement appoint B in Table 3 ? Although the authors mention asymmetry, it is not very clear. The difference is quite significant. Could introduction of artificial/ perturbing the material property or geometry help in capturing this effect numerically? How significant is this deformation in regards to allowing/ blocking the flame and smoke?
Author Response
Response to Reviewer 2 Comments
Thanks for your comments. Here are our comments.
Point 1: Could the authors elaborate on the discrepancy in the displacement appoint B in Table 3.
Response 1: This is an interesting aspect and highlights the need to go in deep of the problem in the full paper. The reason for the discrepancy is that the boundary conditions (e.g. radiation and convection) in the numerical model are considered constant (as indicated in the Eurocode 1), so, being the door symmetric with respect to the x- and z-axes, its thermo-mechanical response is symmetric (displacements equal in points A-D and B-C).
However, in the real test a heat flow directed upwards is generated, hence the temperatures in the upper edge of the door are higher. This effect is the cause of the discrepancy in the measured displacements at point B; conversely the effect of this heat flow is not visible in point A as the hinges constrain displacements. This has been clarified in the text: line 219.
Point 2: Could introduction of artificial/ perturbing the material property or geometry help in capturing this effect numerically?
Response 2: We agree that the origins of this heat flow should be investigated more thoroughly, testing multiple doors and looking for a correlation with the test conditions and doors geometry. It would then be possible to insert this effect in the numerical model by setting for example variable boundary conditions. To calculate the flow using the numerical model, a finite volume model should be used. This work is ongoing as the novel doors are subjected to fire test and we will report on the results. In line with the referees’ suggestion, we have evidenced in the conclusion section that our results are preliminary and there is a strong need of developing a self-consistent FEM methodology that includes the calculation the heat exchange coefficients.
Point 3: How significant is this deformation in regards to allowing/ blocking the flame and smoke?
Response 3: Regarding the effect of this deformation on the door performances, the door considered in this work has passed the FTP fire-test. Accordingly, though these results are considered as preliminary, the fundamental message, i.e. the importance of the thermal bridge in the fire resistance, is fully unaffected by the eventual inaccuracies due to some simplifications in the calculation code, which, by the way, are carried out as suggested by the pertinent Eurocode.
We wish to thank the referee for careful reading and stimulating comments.
Kind regards.
Reviewer 3 Report
In the manuscript entitled “Numerical modeling of fire resistance test as a tool to design light-weight marine fire doors: a preliminary study”, the authors have examined a thermal performance of a fire door made of carbon steel. A numerical simulation was performed, and the results were compared with the experimental data. This study is considered to be important for developing a fire safety technology to prevent spreads of fire and smoke in ships. However, as the authors used the word “preliminary” in the title, the provided data and discussion are not enough to make fire safety science of marine fire doors progress forward. In particular, the manuscript should include conclusion of this study as a scientific article even if the study is preliminary. In addition, unrealistic assumptions that convective heat transfer coefficients were constant were used in this study. Basically, a heat transfer coefficient of a natural convective flow depends on time, location on an onject, and Grashof number. Therefore, the numerical results as shown in Figures 3 are unrealistic.
Based on the above considerations, the reviewer thinks that this paper should not be accepted.
Author Response
Response to Reviewer 3 Comments
Point 1: In the manuscript entitled “Numerical modeling of fire resistance test as a tool to design light-weight marine fire doors: a preliminary study”, the authors have examined a thermal performance of a fire door made of carbon steel. A numerical simulation was performed, and the results were compared with the experimental data. This study is considered to be important for developing a fire safety technology to prevent spreads of fire and smoke in ships. However, as the authors used the word “preliminary” in the title, the provided data and discussion are not enough to make fire safety science of marine fire doors progress forward. In particular, the manuscript should include conclusion of this study as a scientific article even if the study is preliminary. In addition, unrealistic assumptions that convective heat transfer coefficients were constant were used in this study. Basically, a heat transfer coefficient of a natural convective flow depends on time, location on an onject, and Grashof number. Therefore, the numerical results as shown in Figures 3 are unrealistic.
Response 1: Thanks for your comment and we are sorry that you find the paper not publishable.
Whereas we agree with the referee that there is need for a thorough investigation of the phenomena occurring in the fire test and an appropriate assessment of the calculation methodology, which is indeed the central point of our investigation, this point will be addressed in a next paper. The use of preliminary in the title is specifically linked to this aspect and, in fact, we now added a warning in the conclusions concerning the preliminary nature of our results.
It is now highlighted in the conclusion that there is need to develop a self-consistent methodology to calculate the heat exchange coefficients.
We are sorry that the referee did not appreciate the most important aspect of the papers, which is by no way affected by the preliminary nature of this paper, i.e. the assessment of the remarkable role of the thermal bridge at the door edges on the fire-resistance, along with the parameters that allow its reduction. We believe that this message, these considerations and the indications how to construct lighter and thinner fire-doors are important aspects in the weight economy on board ship, although fully compliance with the fire-safety regulations.
As for the formal aspects we wish to recall that have adopted specific constant radiation and convection coefficients, being a preliminary study, as recommended in Eurocode 1.
We wish to thank the referee, though we are sorry that he has not perceived what we believe to be the most important message of our paper, i.e effects of the thermal bridge. We hope that in light of the very helpful referees’ suggestions, the revised version will be appreciated by him as well. Certainly his appreciable comment will be taken into account in the full paper.
Kind regards.
Round 2
Reviewer 1 Report
I have no further comments
Reviewer 3 Report
Although it is questionable for this reviewer that this study has value as a scientific article, the revised paper may be able to be accepted as a technical paper in Journal of Marine Science and Engineering.